# Bayesian Oracle for bounding information gain in neural encoding models

**Konstantin-Klemens Lurz**[1,*], **Mohammad Bashiri**[1,*], **Edgar Y. Walker**[2], **Fabian H. Sinz**[1,3,†]

[1] Institute for Bioinformatics and Medical Informatics, University of Tübingen, Germany

[2] Department of Physiology, Computational Neuroscience Center, University of Washington, USA

[3] Department of Computer Science, University Göttingen, Germany

[*]equal contribution, [†]`sinz@cs.uni-goettingen.de`

## Abstract

In recent years, deep learning models have set new standards in predicting neural population responses. Most of these models currently focus on predicting the mean response of each neuron for a given input. However, neural variability around this mean is not just noise and plays a central role in several theories on neural computation. To capture this variability, we need models that predict full response distributions for a given stimulus. However, to measure the quality of such models, commonly used correlation-based metrics are not sufficient as they mainly care about the mean of the response distribution. An interpretable alternative evaluation metric for likelihood-based models is *Normalized Information Gain* (NInGa) which evaluates the likelihood of a model relative to a lower and upper bound. However, while a lower bound is usually easy to obtain, constructing an upper bound turns out to be challenging for neural recordings with relatively low numbers of repeated trials, high (shared) variability, and sparse responses. In this work, we generalize the jack-knife oracle estimator for the mean—commonly used for correlation metrics—to a flexible Bayesian oracle estimator for NInGa based on posterior predictive distributions. We describe and address the challenges that arise when estimating the lower and upper bounds from small datasets. We then show that our upper bound estimate is data-efficient and robust even in the case of sparse responses and low signal-to-noise ratio. We further provide the derivation of the upper bound estimator for a variety of common distributions including the state-of-the-art zero-inflated mixture models, and relate NInGa to common mean-based metrics. Finally, we use our approach to evaluate such a mixture model resulting in 90% NInGa performance.

## 1 Introduction

In recent years, systems neuroscience has seen great advancements in building neural encoding models of population activity [24; 1; 3; 11; 21; 16; 6; 23]. Most of these models focus on estimating the conditional mean of the response distribution given a stimulus and are consequently evaluated on mean-based measures such as correlation or fraction of explainable variance explained (FEVE). However, neural responses exhibit a great deal of variability even when the animal is presented with the same stimulus. This variability is not just noise, but might be a symptom of underlying neural computations. In fact, many normative theories that link first principles to neural response properties, like the Bayesian brain hypothesis [18], neural sampling [12; 4] or probabilistic population codes [17], make predictions or rely on the variability of neural activity around the mean [15; 13; 5]. If we want to use neural encoding models as a quantitative underpinning for these theories, models are needed which accurately predict and are evaluated on complete response distributions. While progress has been made at building such models [22; 2], it is not clear what upper bound on the performance we can expect. However, this question is important as it gives us an indication how close our models are to the true system.

In the case of mean-predicting models, correlation-based metrics are often used for evaluation [16; 8]. Correlation is an interpretable measure since it is naturally bounded between $-1$ and $1$. However, for vanilla correlation, it is impossible for any model to achieve a correlation of 1 in the presence of

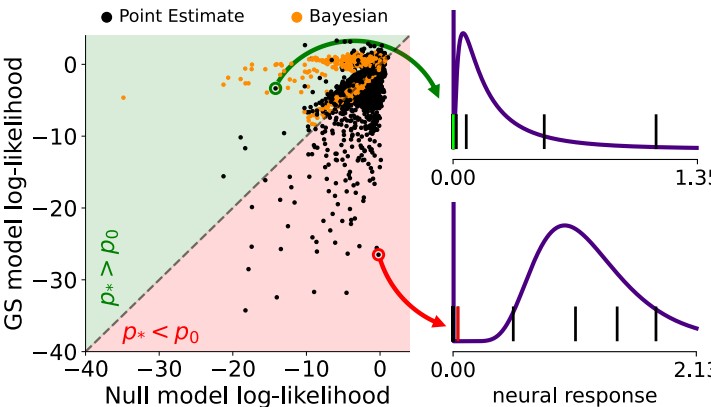

Figure 1: Comparison of lower and upper bound likelihood estimates (Null vs GS) per neuron. **Left:** For many neurons, the PE approach yields worse GS than the Null score. The Bayesian method results in the expected outcome of upper bound scores being higher than lower bound scores. **Right:** Two example neurons demonstrating where the PE method fails (red) or succeeds (green).

trial-to-trial fluctuations. Therefore, model correlation is often normalized by an upper bound oracle estimator [19; 16], which is commonly obtained by computing point estimates of the conditional mean using the responses to repeated presentations of the same stimulus. For a likelihood-based metric, a similar normalization to a bounded and interpretable scale would be desirable, especially for: 1) Assessing whether a model has achieved its "best possible" performance for a given dataset, and 2) comparing models that are trained on different datasets, which can exhibit different levels of achievable performance. To this end, one can use Normalized Information Gain (NInGa) [14], which uses an estimate of both upper and lower bound, to put the likelihood between two meaningful values. However, the challenge lies in how these bounds can be obtained for noisy neural responses.

In this work, we develop a robust way to estimate such lower and upper bounds for NInGa on neuronal responses. We show that a point estimate approach for obtaining the upper bound fails and demonstrate that this is caused by the lack of robustness for the estimate of moments beyond the mean. This is especially pronounced when dealing with data that have few samples, sparse responses, and low signal-to-noise ratios which are common characteristics of neural responses. To mitigate this problem, we propose a generalization of the point estimate approach to a full Bayesian treatment using posterior predictive distributions. Our approach yields lower and upper bounds which are proven to be robust to all the above-mentioned complexities in neural data. We derive a general expression for the Bayesian estimator for zero-inflated distributions that can be efficiently estimated under very general conditions by solving only a single one-dimensional integral on a bounded interval. These distributions capture the sparse nature of neural responses, in particular for 2-photon recordings, and include state-of-the-art zero-inflated mixture models [22; 2]. Using this full-likelihood-based metric, we then evaluate a zero-inflated mixture model and find that it performs remarkably well at 90% NInGa. Finally we experimentally and mathematically relate NInGa to other common metrics for the performance of neural prediction models which are based on the mean and derive general conditions under which likelihood and correlation as a metric identify the same predictive function.

## 2 METHODS

### 2.1 NORMALIZED INFORMATION GAIN AND PROBLEMS WITH POINT ESTIMATES

**Information Gain** Let $p(y|x)$ denote the distribution of a neuron's response $y$ to a stimulus $x$. In order to evaluate and interpret the modeled distribution $\hat{p}(y|x)$ we use Normalized Information Gain (NInGa) [14; 20] which sets the model likelihood on an interpretable scale between an estimated lower and upper bound:

$$\text{NInGa} = \frac{\langle \log \hat{p}(y \mid x)\rangle_{y,x} - \langle \log p_0(y)\rangle_{y,x}}{\langle \log p_*(y \mid x)\rangle_{y,x} - \langle \log p_0(y)\rangle_{y,x}} \tag{1}$$

using a Null distribution $p_0(y)$ and a Gold Standard distribution $p_*(y|x)$. This method of computing NInGa can be interpreted as a normalized comparison of lower bounds of mutual information [20, and Appendix E]. In general, IG is completely flexible in the choice of Null, Gold Standard, and trained model distribution. Therefore, it can be used for model comparison across different distribution families as long as the GS and Null model are kept the same. The *Null model* should reflect basic

aspects of the response. Here, we choose a Null model that does not account for any stimulus-related information, resulting in the marginal distribution of responses $p_0(y)$. The *Gold Standard (GS) model* $p_*(y|x)$, on the other hand, should be the best possible approximation of the true conditional distribution $p(y|x)$. Here, we use an oracle model that has more information than the model under evaluation, such as access to repeated presentations of the same stimulus. Importantly, we do this in a leave-one-out fashion: given a set of $n$ repeats, the GS parameters of a target repeat $i$ are estimated from $n-1$ left-out repeats $\backslash i$. However, as we demonstrate below, estimating a robust GS model can be challenging.

**Point Estimate (PE) GS model**   The parameters $\theta$ of the upper bound estimator can be obtained as point estimates (PE) from $n-1$ left-out repeats:

$$p_*(y_i|\mathbf{y}_{\backslash i}, x) = p(y_i|\theta_i) \quad \text{with } \theta_i = f(\mathbf{y}_{\backslash i}),$$

where $f$ is a function used to obtain the point estimate of $\theta$. In our case, $f$ represents moment matching. For correlation-based performance metrics of neural prediction models, a jack-knifed mean estimator over repeated presentations of the same stimulus $E[y_i|x] = \frac{1}{n-1}\sum_{y_j \in y_{\backslash i}} y_j$ is commonly used as an oracle predictor for the conditional mean to obtain an upper bound on the achievable performance in the presence of noise [19; 16]. While the posterior predictive distribution is generally conditioned on stimulus $x$ because it requires responses to the repeated presentations of the same stimulus, for the remaining of this manuscript we will drop the conditioning on $x$ for brevity.

**Problems with the PE approach**   To demonstrate the problems with point estimate GS models, we modeled neural responses with a zero-inflated Log-Normal likelihood and estimated the upper bound using the PE approach (see Appendix A.1 for details on data, [22; 2] for details on zero-inflated distributions, and Appendix B for the moment matching derivations). Since the GS model estimates parameters per stimulus, it should yield higher likelihood values than the Null model whose parameters are not stimulus-specific. However, applying the PE approach to neural data, we observed that the Null model outperforms the GS model for the majority of neurons (Fig. 1, black points). The reason for this effect is that the PE approach is sensitive to the sparse distribution of the data, which combined with few responses per stimulus results in an overconfident estimation of the GS parameters (see Fig. 1 on the right; Appendix D for a more detailed analysis on where the PE fails).

## 2.2   Bayesian Gold Standard Model

**Gold Standard Model based on posterior predictive distributions**   To avoid an overconfident GS model, we add uncertainty to the parameter estimation and estimate the GS model in a fully Bayesian fashion via the full posterior predictive distribution:

$$p_*(y_i|\mathbf{y}_{\backslash i}) = \int_{-\infty}^{\infty} \underbrace{p(y_i|\theta)}_{\text{likelihood}} \underbrace{p(\theta|\mathbf{y}_{\backslash i})}_{\text{posterior}} \mathrm{d}\theta \tag{2}$$

Note that the PE approach is a special case within this framework for which $p(\theta|\mathbf{y}_{\backslash i}) = \delta(\theta - f(\mathbf{y}_{\backslash i}))$ is collapsed onto a delta distribution.

**Efficient estimation for zero-inflated distributions**   In general, the integral in equation 2 is intractable and the posterior predictive distribution can only be evaluated using numerical approximations. Only for certain choices of likelihood the integral can be solved analytically if the right conjugate prior was chosen.

Here, we show that the posterior predictive distribution can be efficiently computed for the class of zero-inflated distributions [22, Fig. 2]. In principle, such a distribution is a mixture of a delta-distribution at zero and a density of a positive part. In practice, we replace the delta-distribution with a uniform distribution in a small interval $[0, \tau)$ and shift the positive part to the interval $[\tau, \infty)$ so that the two mixture components do not overlap

The probability density for a single stimulus $x$ is then defined as:

$$\hat{p}(y|x) = (1 - q(x)) \cdot \underbrace{p_u(y)}_{\text{uniform}} + q(x) \cdot \underbrace{p(y|\theta_1(x))}_{\text{positive distribution on } [\tau, \infty)}, \tag{3}$$

Figure 2: Graphical model for a zero-inflated distribution. A stimulus $\mathbf{x}$ determines the probability $q$ whether a neurons fires or not and the parameters $\theta_1$ of the response distribution of the non-zero response distribution. A Bernoulli random variable $m_i$ determines whether a neuron fires on a particular trial $i$. If $m_i = 1$ a response is drawn from $p(y_i|\theta_1)$, otherwise from $p(y_i|\theta_0)$.

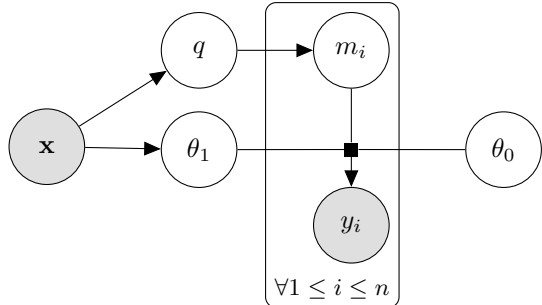

where the mixing proportion $q$ and the parameters $\theta_1$ are dependent on $x$. When using a predictive model, $q$ and $\theta_1$ are typically predicted from the stimulus $x$ using an encoding model. Zero-inflated likelihoods are the basis of current state-of-the-art likelihood-based neural encoding models [22; 2].

Here we show that the posterior predictive distribution for a zero-inflated likelihood boils down to a one dimensional integral over $q$ if the posterior predictive distribution of the positive part is known.

**Lemma 1.** *The posterior predictive distribution of a zero-inflated distribution as defined in equation 3 is given by*

$$p(y_i|\mathbf{y}_{\setminus i}) = \begin{cases} p(y_i|\mathbf{y}_{\setminus i}^0) \cdot \int_q (1-q) \cdot p(q|\mathbf{y}_{\setminus i})\,dq & \text{if } y_i < \tau \\ p(y_i|\mathbf{y}_{\setminus i}^1) \cdot \int_q q \cdot p(q|\mathbf{y}_{\setminus i})\,dq & \text{if } y_i \geq \tau \end{cases}$$

*where $\mathbf{y}_{\setminus i}^0$ and $\mathbf{y}_{\setminus i}^1$ denote the set of zero and non-zero responses in $\mathbf{y}_{\setminus i}$, respectively.*

*Proof.* See Appendix C. □

This means that the posterior predictive distribution can be computed efficiently for a large class of positive distributions, including the zero-inflated Log-Normal or even zero-inflated Flow models [2]. In the next section, we demonstrate that this Bayesian treatment yields a GS model that is more robust against outliers and yields higher likelihoods than the Null model, as expected (Fig. 1, orange points).

## 3 EXPERIMENTS

### 3.1 ANALYSIS OF GOLD STANDARD MODELS

In this section, we investigate why the Bayesian approach outperforms the PE and test its robustness under different numbers of left-out repeats $\mathbf{y}_{\setminus i}$ and different signal-to-noise ratios. For this analysis, we used responses from 7672 neurons to 360 stimuli, each repeated 20 times as well as simulated data. Details about the datasets are provided in Appendix A.1 and Appendix A.2, respectively. We base our posterior predictive distribution on a zero-inflated Log-Normal distribution with zero-threshold $\tau = \exp(-10)$ and parameters $\theta_1 = (\mu, \sigma^2)$, referring to the first and second moment of the log-transformed positive responses (Appendix C). The conjugate prior for the Log-Normal part of the distribution is a Normal-Inverse-Gamma distribution $p(\theta_1) = \mathcal{N}G^{-1}(\mu, \lambda, \alpha, \beta)$ in log space whose parameters can be chosen freely. We discuss choices of prior parameters later in section 3.2.

**Bayesian GS estimates higher order moments better** In order to determine which parameters of the likelihood profits most from the Bayesian estimation, we compared GS models where the individual parameters are either estimated via the PE or the Bayesian approach (Fig. 3**a**). For this we used the largest number of left-out repeats $n - 1 = 19$ available in our real neuron dataset. First, we observe that the likelihood improves with the Bayesian estimation of $\mu$ (orange vs. yellow bar) as well as $\sigma^2$ (light blue vs. yellow bar) individually. Consequently, the highest performance is achieved when both parameters are estimated via the Bayesian approach (dark blue vs. yellow bar). Interestingly, the relative gain in performance is much higher for $\sigma^2$ than for the $\mu$, reflecting a lower robustness of the higher moments compared to the first moment in log space.

**Bayesian GS is data-efficient** Datasets can vary in how many repeats per stimulus they contain. Since a metric should be comparable across datasets, NInGa ought to yield consistent estimates for

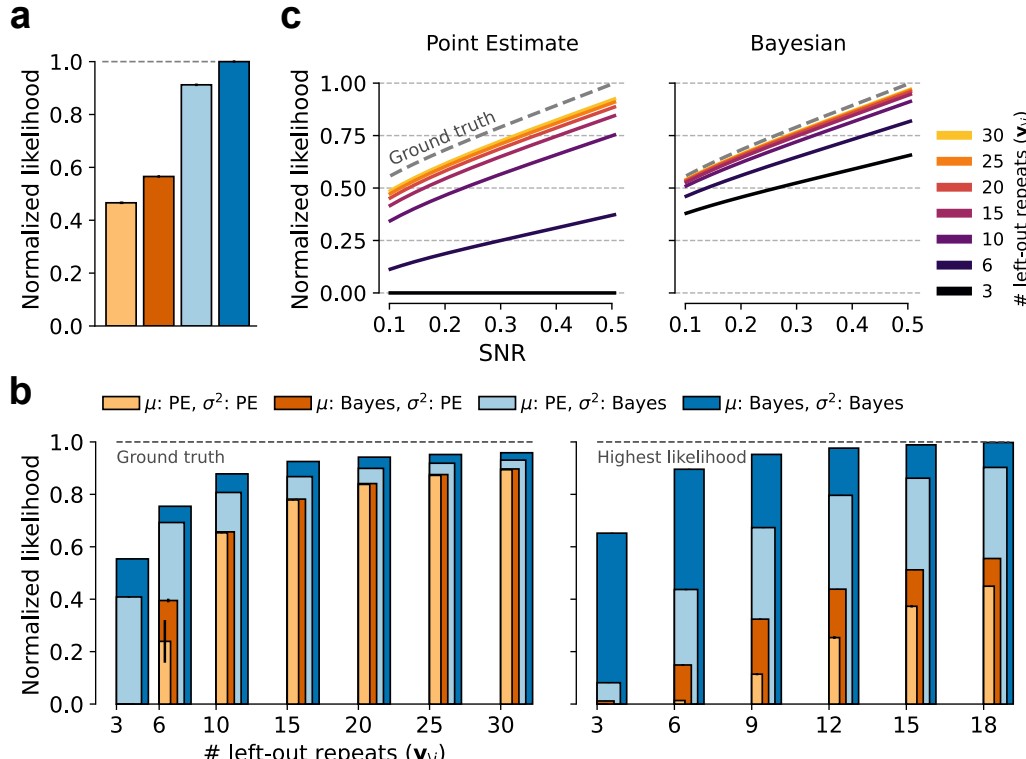

Figure 3: Comparison of the point estimate and Bayesian GS model. **a:** Comparison of different GS models where the individual parameters are either estimated via the PE or the Bayesian approach. The number of left-out repeats $\mathbf{y}_{\setminus i}$ is 19. Colors are the same as in **b**. Normalized wrt. the max. likelihood value, i.e. the dark blue bar. **b:** Similar to **a** but for different numbers of left-out repeats $\mathbf{y}_{\setminus i}$. **Left:** Simulated data. The likelihood is normalized w.r.t. the ground truth likelihood. **Right:** Neural responses. Likelihood is normalized w.r.t. the maximum likelihood value, i.e. the dark blue bar at 18 left-out repeats. **c:** Upper bound likelihood scores for different signal-to-noise ratios and different number of left-out repeats $\mathbf{y}_{\setminus i}$. **Left:** Point Estimate. **Right:** Bayesian. Likelihood is normalized w.r.t. the ground truth likelihood. In all panels the likelihood values are averaged over stimuli and neurons, and the error bars and shaded areas show SEM over 5 random selections of the left-out repeats.

different numbers of left-out repeats $\mathbf{y}_{\setminus i}$, in particular in the regime of low $n-1$. The right panel of Fig. 3**b** shows this on neural data where we first observe that the results of Fig. 3**a** are qualitatively consistent for different numbers of repeats. The effect of the Bayesian parameter estimation on the likelihood performance, however, is much more pronounced in the low $n-1$ regime: As the number of left-out repeats decreases the PE approach suffers much more than the Bayesian and it completely fails at $n=3$ (vanishing yellow bar). To test the higher $n-1$ regime, we simulated neural responses since the real neural dataset contained maximally 20 repeats. In the left panel of Fig. 3**b** we explored the differences between the two approaches for up to 30 repeats and observed that the Bayesian treatment consistently yields a better likelihood than the PE estimate (yellow bar does not completely converge to dark blue bar). The probabilistic treatment of $\mu$, however, seems to become less important in the high $n-1$ regime than that of $\sigma^2$, reflecting the higher robustness of the first moment compared to the second moment in log space (compare the difference between orange and yellow for high vs. low $n-1$).

**Bayesian GS is robust to different SNRs** Apart from different numbers of repeats, datasets can also vary in terms of signal-to-noise ratio. We therefore simulated neural data with different underlying means and variances per stimulus, resulting in different SNR values (see Appendix A.2 for details). We then tested Bayesian and point estimate GS models on this data (Fig. 3 **c**) and observed that the Bayesian approach consistently outperforms the PE approach across all SNRs (data for Fig. 3**b** left panel had an SNR of $0.42$).

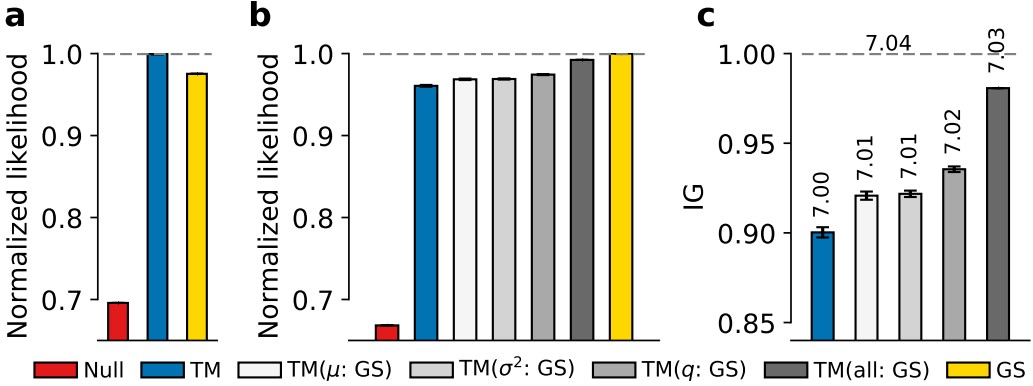

Figure 4: Real neural data evaluation of a neural encoding model trained on a zero-inflated Log-Normal likelihood (ZIL) using the Bayesian upper bound estimate (GS). **a:** A sub-optimal prior for the GS model results in a lower performance than the trained model (TM). Normalized wrt. the max. likelihood value, i.e. the blue bar. **b:** The GS model outperforms the TM when its prior was optimized. The TM estimates all parameters similarly well. Grey scale colors indicate models from the same distribution as the TM (ZIL) for which all parameters are estimated by the trained model except for the parameters in parentheses. Normalized wrt. the max. likelihood value, i.e. the yellow bar. **c:** Normalized Information Gain (NInGa) for the TM and the models grey scale models from **b**. Values on top of bars indicate the likelihood per image and per neuron in bits. In all panels the likelihood values are averaged over stimuli and neurons, and the error bars and shaded areas show SEM over 5 random initializations of the TM weights. There are no errorbars on the red, yellow and dark grey bars since they do not involve a TM.

## 3.2 EVALUATING NEURAL ENCODING MODELS VIA NINGA

In Section 3.1 we demonstrated that the Bayesian approach for obtaining an upper bound estimator greatly outperforms the point estimate (PE) approach in several aspects. From now on, we therefore only use the Bayesian estimator when referring to the Gold Standard Model. In this section, we use the lower and upper bound estimates to evaluate neural encoding models via Normalized Information Gain.

To this end, we trained an encoding model on a dataset which is publicly available [16] containing the responses of 5335 neurons in mouse primary visual cortex to 5094 unique natural images. The dataset was split into 4472 stimulus-response pairs for training, 522 for validation, and 100 for testing. The stimuli in the test set were each repeated 10 times resulting in $n - 1 = 9$ repeats for the GS model to be computed on. The neural encoding model and the training are the same as in the model provided by Lurz et al. [16]. Briefly, the encoding model consists of two parts: (1) A core network which is shared across neurons with four convolutional layers (some of them depth-separable [9]) resulting in 64 feature channels, followed by batch normalization and ELU nonlinearity. And (2) a neuron-specific Gaussian readout mechanism [16] that learns the position of the neuron's receptive field (RF) and computes a weighted sum of the features at this position along the channel dimension. While Lurz et al. [16] used Poisson loss to train the models, we chose the negative loglikelihood of a zero-inflated Log-Normal (ZIL) distribution as a loss function. This means that the model needs to predict three parameters ($q$, $\mu$, $\sigma^2$) instead of a single mean firing rate $\lambda$ as in Lurz et al. [16]. The readout thus learns three weight vectors to combine features extracted via the core network, at neuron's learned RF position. Since the metric that the model will be evaluated on is NInGa, the early-stopping criterion was changed from correlation to likelihood. The results of these experiments with neural encoding models are summarized in Fig. 4 and will be explained in detail below.

**Choice of prior is crucial for the GS model** Up to this point, we chose the prior hyper-parameters neuron-independently: We chose the parameters of the Normal-Inverse-Gamma prior based on the average conditional mean $\mathrm{E}_x[\mathrm{E}_y[y|x]]$ and average conditional variance $\mathrm{E}_x[\mathrm{Var}_y[y|x]]$ of our dataset. This left us with number of neurons samples which we fit the prior on, resulting in one identical prior for every neuron. However, we observed that the resulting GS model (yellow bar) was outperformed by the trained encoding model (TM model, blue bar) and did not provide a good estimate of a

performance upper bound (Fig. 4 **a**). To obtain a better upper bound oracle model, we therefore optimized the prior hyper-parameters directly on the sum of leave-one-out GS likelihoods via gradient descent for each single neuron. This is analogous to other oracle models that have access to more information than the predictive model under evaluation, thus providing an upper bound. Note that this approach results in a more conservative estimate of the model performance (Fig. 4 **b**, compare the blue and yellow bars). We also tried other approaches (e.g. MAP estimate) to obtain a better GS model but none of them outperformed the Posterior Predictive GS (see Appendix I).

**Encoding model captures all parameters similarly well**   In order to investigate which parameters of the response distribution the encoding model predicts well, we conducted an experiment similar to the one shown in Fig. 3 **a**: We compared the likelihood of the trained model (TM, blue) to cases where we matched one or all of its three parameters ($q$, $\mu$, $\sigma^2$) to the GS model (see Fig. 4 **b** blue bar vs. grey bars). While we could match the parameter $q$ directly, we used moment matching to obtain the parameters of the non-zero part of the trained model (Log-Normal distribution) from the non-zero part of the GS model (Log-Student-t). We observed that the likelihood of the trained model improved when each of the three parameters were matched with the GS model individually (three lighter grey bars vs. blue bar), where $q$ yielded the highest increase. Matching all three parameters, however, did not result in the same performance as the GS model itself (darkest grey vs. yellow). Since all parameters of the ZIL model were fitted on the GS model, the reason must be the difference in distributional shape of the positive part: Log-Normal for ZIL vs. Log-Student-t for the GS model.

**Encoding model performance is at 90% NInGa**   The final NInGa value of the trained model using the Null model and the GS model with optimized prior hyper-parameters can be seen in Fig. 4 **c**. It performs remarkably well at 90% NInGa (blue bar), which corresponds to a likelihood of 7.00 bits per image and neuron (printed value above the bar). The effect of the parameter matching (grey bars) is more emphasized and suggests that the largest performance gain can be achieved in future models by improving the prediction of the parameter $q$. We performed additional analyses on multiple datasets to show how NInGa facilitates model comparison across different datasets (see Appendix J).

## 4   RELATION BETWEEN NINGA AND OTHER METRICS

Neural predicting models have so far been mostly evaluated using metrics such as fraction of explainable variance explained (FEVE) [7], correlation, and fraction oracle [19; 16]. While we cannot expect that there is a one-to-one relationship between NInGa and these metrics, as NInGa is sensitive to the entire response distribution whereas the commonly used metrics mostly focus on the mean response of a neuron for a given stimulus, we can nevertheless relate NInGa to these metric under certain assumptions. In this section we provide a summary of the relationships to these metrics. Details and proofs can be found in Appendices F and G.

Generally, we will demonstrate the relations in two steps: First we show that NInGa linearly depends on the expected Kullback-Leibler (KL) divergence between the true distribution $p(y|x)$ and the model distribution $\hat{p}(y|x)$. Then we derive how this KL divergence relates to other metrics.

**NInGa is a linear function of** $\langle D_{KL}[p(y|x), \hat{p}(y|x)]\rangle_x$   The NInGa described in Eq. 1 can be re-written in terms of KL divergences between estimated/fitted distributions and the true conditional distribution $p(y|x)$ (see Appendix E for the detailed derivation):

$$NInGa = \frac{\langle D_{KL}[p(y|x)||p_0(y)]\rangle_x - \langle D_{KL}[p(y|x)||\hat{p}(y|x)]\rangle_x}{\langle D_{KL}[p(y|x)||p_0(y)]\rangle_x - \langle D_{KL}[p(y|x)||p_*(y|x)]\rangle_x} \qquad (4)$$

This shows that the Normalized Information Gain is a linear function of $\langle D_{KL}[p(y|x)||\hat{p}(y|x)]\rangle_x$ with a negative slope. It is maximized when average Kullback-Leibler divergence between the true distribution and the model distribution is minimized.

**NInGa vs. FEVE**   Using this fact, we now show that NInGa is linear in another commonly used metric, fraction of explainable variance explained (FEVE), for a Gaussian likelihood. For a fixed model noise variance $\hat{\sigma}_\epsilon^2$ the KL-divergence $\langle D_{KL}[p(y|x)||\hat{p}(y|x)]\rangle_x$ is a linear function of FEVE:

$$\langle D_{KL}[p(y|x)||\hat{p}(y|x)]\rangle_x = f(\hat{\sigma}_\epsilon) + \frac{1}{2}(1 - FEVE) \times \frac{\sigma_s^2}{\hat{\sigma}_\epsilon^2}, \qquad (5)$$

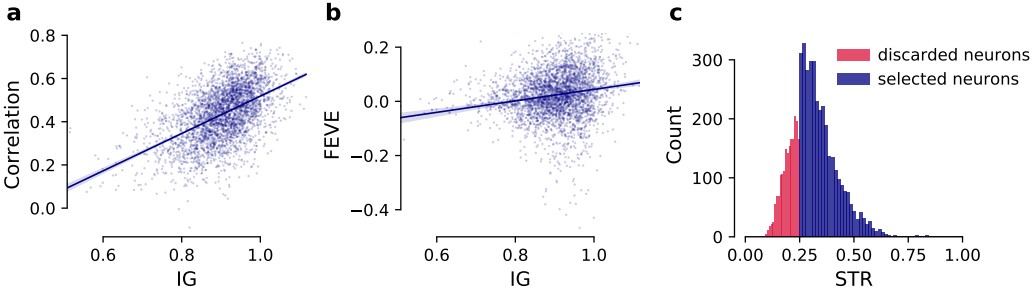

Figure 5: Per neuron comparison of NInGa with correlation and FEVE. The data comes from the TM in Fig. 4 (one of the 5 models from different initializations). **a:** Correlation vs. NInGa. **b:** FEVE vs. NInGa. **c:** The neurons displayed in panel **a** and **b** were selected based on whether their signal-to-total-variance ratio (STR) was above $0.25$. The blue lines depict linear regressions with $95\%$ confidence intervals.

where $f(\hat{\sigma}_\epsilon) = \log\left(\frac{\hat{\sigma}_\epsilon}{\sigma_\epsilon}\right) - \frac{1}{2} + \frac{\sigma_\epsilon^2}{2\hat{\sigma}_\epsilon^2}$ and $\sigma_s^2$ is the signal variance. Since $\langle D_{KL}[p(y|x)||\hat{p}(y|x)]\rangle_x$ and NInGa are also linearly related, then NInGa too is a linear function of FEVE. Note that when the estimated noise variance is the same as the true noise variance, then Eq. 5 becomes:

$$\langle D_{KL}[p(y|x)||\hat{p}(y|x)]\rangle_x = \frac{1}{2}(1 - FEVE) \times SNR$$

See Appendix F for detailed derivations. If the estimated noise variance $\hat{\sigma}_\epsilon^2$ is not fixed, as is the general case, this trend will still be true but the linear relationship between the NInGa and FEVE is more noisy. In Fig. 5**b**, we show that by evaluating the trained model from Fig. 4 on NInGa and FEVE per neuron. Note that we only display neurons whose signal-to-total-variance ratio $STR = \sigma_s^2/\sigma_y^2 = \sigma_s^2/(\sigma_s^2 + \sigma_\epsilon^2)$ exceeds a threshold which we set to $0.25$ (see Fig. 5**c**).

**NInGa vs. correlation** Another commonly used metric is the correlation $\rho(\hat{\mu}_x, \mu_x)$ between the model prediction $\hat{\mu}_x = \langle \hat{y} \rangle_{\hat{y}|x}$ and trial averaged responses $\mu_x = \langle y \rangle_{y|x}$. Assuming a Gaussian likelihood, like in the relation to FEVE, we show that for a fixed model noise and signal variance $\langle D_{KL}[p(y|x)||\hat{p}(y|x)]\rangle_x$ is a linear function of the correlation between $\mu_x$ and $\hat{\mu}_x$:

$$\langle D_{KL}[p(y|x)||\hat{p}(y|x)]\rangle_x = f(\hat{\sigma}_\epsilon) + \frac{1}{2}\left(1 + \frac{\hat{\sigma}_s^2}{\sigma_s^2} - \frac{2\hat{\sigma}_s}{\sigma_s}\rho(\hat{\mu}_x, \mu_x)\right) \times \frac{\sigma_s^2}{\hat{\sigma}_\epsilon^2} \tag{6}$$

As before, if the model noise variance matches the true noise variance, $\sigma_\epsilon^2 = \hat{\sigma}_\epsilon^2$, we have:

$$\langle D_{KL}[p(y|x)||\hat{p}(y|x)]\rangle_x = \frac{1}{2}\left(1 + \frac{\hat{\sigma}_s^2}{\sigma_s^2} - \frac{2\hat{\sigma}_s}{\sigma_s}\rho(\hat{\mu}_x, \mu_x)\right) \times SNR$$

Assuming further that the model's signal variance matches the true signal variance, $\hat{\sigma}_s^2 = \sigma_s^2$, we get:

$$\langle D_{KL}[p(y|x)||\hat{p}(y|x)]\rangle_x = (1 - \rho(\hat{\mu}_x, \mu_x)) \times SNR$$

It is worth noting that the assumptions for having a linear relationship with KL divergence are stronger in the case of correlation than that of the FEVE. Specifically, there is an additional dependence on the signal variance in the case of correlation. This makes sense as FEVE is inherently sensitive to signal variance while correlation is not (see Appendix G for the detailed derivation). As in the case of FEVE, we empirically show (Fig. 5 **a**) the relation between NInGa and correlation on our trained model from Fig. 4. While the general linear trend can be observed, this relation is very noisy, as expected. Note again, that we only show neurons that pass the threshold of $STR \geq 0.25$.

**Does a high likelihood always correspond to a high correlation?** Correlation only reflects good estimation of the mean of a distribution (see Appendix H). NInGa, on the other hand is a likelihood-based metric and as such depends on the correct estimation of all parameters of the likelihood. So does a high likelihood correspond to a high correlation? The answer is: it depends.

Correlation is high if the conditional mean is accurately estimated. However, since the mean of the distribution is in general not directly a parameter of the likelihood it is not necessarily estimated

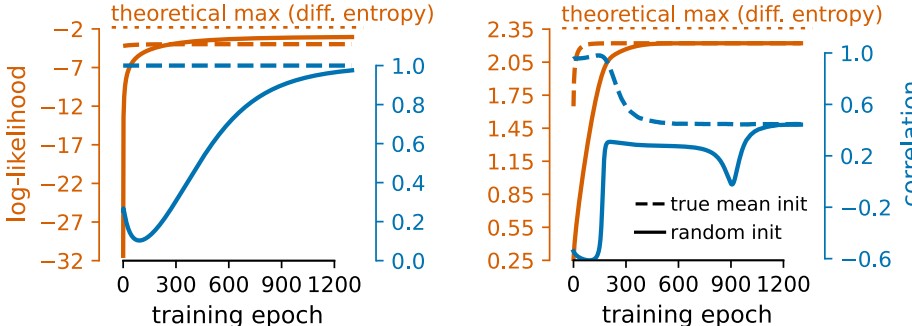

Figure 6: Comparison of log-likelihood and correlation of a model which is optimized via maximum likelihood estimation. **Left:** The model distribution is from the exponential family and has the mean as a sufficient statistic. The optima of the two metrics coincide but an improvement in likelihood does not necessarily mean an improvement in correlation. Data distribution: Normal. Model distribution: Gamma. **Right:** The model distribution does *not* fulfill these criteria. No similar behavior of likelihood and correlation can be guaranteed. If initialized with optimal correlation, the correlation after training has decreased (dotted line). Data distribution: Gamma. Model distribution: Chi-squared.

accurately even if optimal likelihood is achieved. Therefore, there is in general no guarantee that a model that outperforms another model on likelihood also outperforms this other model on correlation. However, if the distribution is 1) a member of the exponential family and 2) has the mean as a sufficient statistic [10], an optimal likelihood also implies an optimal correlation. A distribution that fulfills these two criteria guarantees that for optimal likelihood, the mean is optimally estimated and with it the correlation. We demonstrate this effect using simulated data (see Appendix A.2) in Fig. 6 in the left panel: A model of such a distribution is being optimized on toy data via maximum likelihood estimation and at the end of training the optima of likelihood and correlation coincide. Note however that during training, i.e. for non-optimal likelihood, a training step which improves likelihood does not necessarily yield an improvement of correlation (compare decreasing blue line vs. increasing orange line). If the distribution does not fulfill the second criterion, i.e. if the mean is not a sufficient statistic, no relation between the correlation and likelihood can be guaranteed, not even at the optimum. This can be seen in Fig. 6 in the right panel: An improvement in likelihood (orange) does not guarantee an improvement in correlation (blue). If the model was initialized such that the correlation is optimal (dotted lines), the maximum likelihood estimation will even result in a model that has lower correlation than it had at the time of initialization (dotted blue line at epoch 0 vs 1200).

## 5 CONCLUSION

In this work, we discussed the challenges in obtaining the lower and upper bound estimates for likelihood-based measures like Normalized Information Gain (NInGa) in order to put the performance on an interpretable scale. We introduced a robust way of obtaining such an upper bound by using the Bayesian framework of posterior predictive distributions. Equipped with this metric, we showed that current neural encoding models are able to predict full response distributions to up to $90\%$ NInGa and we examined which parameters of the distribution the model still fails at predicting. We also gave a detailed derivation for obtaining upper bound estimates for the state-of-the-art family of distributions, the zero-inflated distributions. Finally, we related likelihood-based metrics like NInGa to other metrics which are commonly used in Computational Neuroscience like correlation and FEVE.

There are, of course, also some limitations to the current work. While we were able to fix the catastrophic failure of naive PE estimates (Fig. 1), our estimates of NInGa or the GS model are not perfect as some neurons yield NInGa $> 1$ (Fig. 5) which is not mathematically impossible but would ideally not happen. Better NInGa or GS estimators will mitigate this issue. Finally, our approach is for single neuron likelihoods. However, the neural variation is structured on a population level, for instance through brain states. Since the general approach of NInGa still works in that situation, our framework is flexible enough for a robust GS model of full populations to be derived in future work.

ACKNOWLEDGMENTS

We thank all reviewers for their constructive and thoughtful feedback.

Konstantin-Klemens Lurz is funded by the German Federal Ministry of Education and Research through the Tübingen AI Center (FKZ: 01IS18039A). Mohammad Bashiri is supported by the International Max Planck Research School for Intelligent Systems. Fabian H. Sinz is supported by the Carl-Zeiss-Stiftung and acknowledges the support of the DFG Cluster of Excellence "Machine Learning – New Perspectives for Science", EXC 2064/1, project number 390727645. This work was supported by the Intelligence Advanced Research Projects Activity (IARPA) via Department of Interior/Interior Business Center (DoI/IBC) contract number D16PC00003. The U.S. Government is authorized to reproduce and distribute reprints for Governmental purposes notwithstanding any copyright annotation thereon. Disclaimer: The views and conclusions contained herein are those of the authors and should not be interpreted as necessarily representing the official policies or endorsements, either expressed or implied, of IARPA, DoI/IBC, or the U.S. Government.

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
