# OpenReview forum: "Bayesian Oracle for bounding information gain in neural encoding models"
_ICLR.cc/2023/Conference — ICLR 2023 poster_

### Official Review · Reviewer_1J5x · 2022-10-24

**Confidence:** 3
**Correctness:** 3
**Technical Novelty And Significance:** 2
**Empirical Novelty And Significance:** 2
**Recommendation:** 6

**Clarity, Quality, Novelty And Reproducibility:**

The notation and concepts are confusing in a few places. On the bottom of page 2 the authors say $p_0(y \mid x) = \hat{p}(y)$. However, this seems wrong as one would want the null model to be independent of the choice of $\hat{p}(y)$. That is, suppose I wanted to compare a Poisson encoding model (like Lurz et al) versus a zero-inflated log-normal model. This would seem to imply two different null models in the way the paper is currently written, but I would need to use a common null model to compare ZIL to Poisson in terms of information gain.

I also found the notation $p(y)$ in equation 3 to be confusing. I would write this instead as $p(y \mid x)$ (or is it $\hat{p}(y \mid x)$ here?). The authors say below the equation that $\theta_1$ and $q$ are dependent on $x$. This helps, so ultimately I understand what they are going for here but the notation should be improved so that readers can refer to equations in a more standalone fashion. I think equation 2 and the equation under the point estimate model in section 2.1 similarly are conditioned on $x$ somehow, so a similar revision of notation there would be helpful.

Finally, and most importantly, I would like the authors to include a more detailed description of the Lutz et al model within the main text of this paper and also expand their discussion of this model in the Appendix. In short, I do not feel the paper is currently "self-contained" with all of these details (e.g. it is confusing what is meant by "core network").

**Strength And Weaknesses:**

The main strength of this paper is that it identifies a rather serious deficiency within the current state of the field. Having pointing out this deficiency, it presents a very well-known and rigorous solution.

The main weakness of this paper is that it focuses more-or-less entirely on conceptual first principles. To be sure, these first principles are really important for us to get right, but I would have liked to see more empirical demonstrations that compare multiple encoding models and revisit old results (e.g. Yamins & Dicarlo-style regressions between deep nets trained not on encoding task, but on different image tasks). A skeptical reader may come away unconvinced without a deeper empirical demonstration.

Another weakness of this paper, in my view, is that it spends a long time trying to identify a "Gold Standard" upper bound model, which isn't really necessary for us to compare and rank different encoding models. Thus, a lot of the paper ends up being somewhat uninteresting to me. Further, while it would be nice to have an upper bound on model performance, this isn't easily done. The authors find out in Figure 5a that their encoding model actually out-competes their gold standard, and they then amend their gold standard to be a "true upper bound" in Figure 5b. However, I challenge the claim that this is a true upper bound -- the gold standard model is a parametric model (albiet a pretty good one!) and there is nothing saying someone else couldn't come along with something else to beat it. This is especially true since the gold standard model as currently formulated doesn't share any information across different image types / categories. Presumably this is why it is possible for the deep net encoding model to outperform the gold standard. Overall, I would suggest that the author's revise the paper to modify their language --- i.e. treat the gold standard as a useful benchmark or point of comparison, but not a end-all-be-all upper bound.

A final weakness of this paper is the clarity of the notation and deep net experiments, described below. I believe however that this weakness is addressable in review.

**Summary Of The Paper:**

The author's propose a new metric for evaluating / comparing neural encoding models. They call this metric the Information Gain (IG):

$
\frac{ \langle \log \hat{p}(y \mid x) \rangle  -  \langle{\log p_0(y)} }{ \langle \log p_*(y \mid x) \rangle  - \langle \log p_0(y) \rangle }
$

I would instead suggest calling this something like *Normalized Information Gain* as previous papers (e.g. Kummerer et al, 2015) call the numerator the information gain (and in that case it has the right units of bits/nats).
The IG is a very well-known metric used to compare different models.
Personally, I would prefer to use the term log Bayes factor or log-likelihood ratio, but this is just a matter of taste/background.

Much of the paper is spent trying to find a good "Gold Standard" model $p_{\star}(y \mid x)$, which is meant to upper bound the performance of any model. The authors use a reasonable Empirical Bayes methodology here. One concern is that they find encoding models that outperform this "Gold Standard." This is not too much of a problem in my view, but the authors may try the following simple trick/suggestion &mdash; I would try taking a having $p_{\star}(\cdot)$ be a weighted average of the current gold standard model $p(y_i \mid \mathbf{y}\_{\backslash i} , x_i )$ and the *global* distribution over all images (not just those with the same image input as trial $i$). The idea is that one has an extremely large number of trials so you could get a very accurate kernel density estimate of $p(y)$ and shrink your under-powered parametric estimator of $p_*(\cdot)$ slightly towards this (biased, stimulus-independent) of  the neural response distribution. I'm not sure this will work but it may be worth trying.

The authors conclude by relating IG to existing metrics within the field like correlation and fraction of explainable variance.

**Summary Of The Review:**

I have to admit this paper makes me somewhat depressed. To me it is blatantly obvious that information gain is a superior measure of performance relative to existing conventions. It doesn't feel like we should require a whole paper to set the field straight. But sadly I think this may be a necessary story for people to hear. Thus, I do see merit in this paper.

I am so far proposing a borderline reject to this paper because of the missing details and unclear notation in areas outlined above. However, I'm open to raising my score if the authors revise the paper to provide those missing details.

---

> ### Author Response · Authors · 2022-11-15
> **Authors response to Reviewer 1J5x (part 2)**
>
> **Re: Notation fixes**: Thanks for pointing these out. We provide some explanation about the reasons for choosing the notation in the paper and explain how we have changed the paper to address these issues.
>
> - **Not treating the GS as an end-all-be-all upper bound**: This is, of course, not the level of generality we intended to claim. It is possible to achieve higher GS performance in principle by other methods or better choices of distribution. Thanks for catching this inaccuracy in our description. In section 3.2 we changed "To obtain a true upper bound oracle model, we [...]" to "To obtain a better upper bound oracle model, we [...]".
>
> - **$p_0(y∣x)=\hat{p}(y)$**: Thanks for pointing this out. One can use IG to compare different model classes.  As you pointed out, there is a need for a common GS and a common Null. We changed the manuscript accordingly to keep it generic and use $p_0(y)$.
>
> - **$p(y)$ in equation 3**: You are right, and in fact, the correct notation for Eq. 3 would be $\hat{p}(y|x)$. We fixed this in the paper. Regarding the posterior predictive (Eq. 2), there is also a need for conditioning. We added it to Eq. 2 but also mentioned that we will drop it for the rest of the manuscript for brevity.
>
> - **paper is not "self-contained"**: We added more details about the Lurz model in the paper.

---

> ### Author Response · Authors · 2022-11-15
> **Authors response to Reviewer 1J5x (part 1)**
>
> Thanks a lot for your constructive feedback. We are confident that we can address your concerns through additional analyses as well as the further explanation provided below:
>
> **Re: Naming of the method**: We agree that *Normalized Information Gain (NInGa)* is a more fitting name for Eq. 1. We changed the name in the updated version of the paper. However, we kept the original name in the rebuttal to avoid confusion.
>
> **Re: a mixture model as the GS model**: Thanks for this interesting suggestion. We implemented a GS model, based on your suggestion, as a mixture between the Bayesian GS and the Null model. Our results (Fig. S2) show that the mixture model does not improve beyond the GS model obtained via the Posterior Predictive approach. The mixture model was implemented as a weighted sum of the Bayesian GS and the Null $w_i\cdot p_*(y_i|\mathbf{y}_{\setminus{i}}) + (1-w_i)\cdot p_0(y)$ with $w_i \in [0, 1]$. We optimized $w$ in a leave-one-out manner: we obtained a $w$ for each target repeat (per neuron per image) by optimizing the $w$ on the other repeats (i.e. responses to other presentations of the same stimulus). The complete description of the mixture model and the corresponding results are added to the manuscript under Appendix I.2.
>
> **Re: Too focused on Gold Standard and lack of encoding model comparison**: We agree that using IG for comparing different encoding models would be insightful and interesting. However, this was not the purpose of the current paper. In this paper, we focused on promoting IG as an additional, more holistic, measure for quantifying the performance of neural encoding models, specifically for models that predict a response distribution and not a single value. To additionally obtain an upper bound that helps us understand whether the model has achieved its best possible performance for a given data, IG requires the estimation of a performance upper bound (the GS model). This model is different from the other models because it has access to additional information such as measures of responses to repeated trials (it’s an oracle model). One of the reasons why correlation and $R^2$ are so common as evaluation measures is that they have clear upper bounds. But even for those measures, the upper limit cannot be reached in the presence of noise. This is why they are often corrected by a noise ceiling, for instance, estimated by an oracle (GS) model. Thus our approach parallels common practice in correlation-based metrics.
>
> While a GS model is not a “true upper bound” it nevertheless allows us to put likelihood values into perspective. By using an oracle model that has a bit more information than the encoding model (i.e. access to repeated trials) we believe that we can get reasonable upper-bound estimates. However, for reasons mentioned in the paper, it turned out quite challenging to come up with a good GS model for neural responses, and this is why we focussed the paper on it. One reason is related to the data: neural responses are often sparse and, due to experimental limitations, only a few responses to repeated presentations of the same stimulus are recorded. We addressed this using the posterior predictive framework. However, using a posterior predictive distribution introduced a new challenge for the GS model, which our paper addresses: Deriving the GS model for the current state-of-the-art neural encoding models (Appendix C). All of these are novel and relevant contributions to the field because not only do we promote the likelihood-based metric, but also provide derivations and examples of 1) how state-of-the-art models can be evaluated using IG, and 2) how IG compares to other commonly used metrics and under what assumptions they correspond to one another.

---

> ### Comment · Reviewer_1J5x · 2022-11-25
> **Raising to a borderline accept in response to notational fixes**
>
> I thank the authors for their revisions, which satisfy my major concerns. I am raising my score to boderline accept.
>
> The technical and empirical novelty of this paper is limited, but as I wrote in my original summary I do see some merit in having this message be seen by the community of researchers using deep network features to predict biological responses.

---

### Official Review · Reviewer_ALKJ · 2022-10-25

**Confidence:** 2
**Clarity, Quality, Novelty And Reproducibility:** The paper is clearly written.
**Correctness:** 3
**Technical Novelty And Significance:** 2
**Empirical Novelty And Significance:** 2
**Recommendation:** 6

**Strength And Weaknesses:**

Strength:
The paper is clearly written.
The authors are discussing about an important question: how to evaluate a neural encoding model.
The authors have shown both theoretical derivations and empirical data studies.

Weakness and questions:
The authors mainly test on 1 real dataset, just wonder if the main result (e.g. 90% IG) still holds for other datasets?
The mixture model achieves 90% IG on the real data example. Is this good enough? Can IG be used to compare encoding models with different distribution family?
There are also other common ways to measure the goodness of a neural encoding model, e.g. decoding performance. I wonder if the authors have done comparisons with other measures? Is IG consistent with other measures?

**Summary Of The Paper:**

The authors proposed a method to obtain upper bound for IG. They showed that this method is efficient and robust using simulated data. They derived the estimator on zero-inflated distribution family. Finally they showed that the encoding model achieved high IG performance on a real dataset.

**Summary Of The Review:**

This paper is clearly written and the discussed question is important. But the paper is lack of more real examples and comparisons. The authors only looked at one real dataset and didn't compare with other methods. I have proposed some detailed questions and weakness in the above section.

---

> ### Author Response · Authors · 2022-11-15
> **Authors response to Reviewer ALKJ**
>
> Thank you for your feedback. Below we provide some explanations to address your questions and describe how we have improved (or plan to improve in the coming days) the manuscript accordingly (the manuscript is already updated and uploaded):
>
> **Re: Test IG on more datasets**: While we addressed all other comments in your review, we are still working on producing new results to asses IG for models trained on additional datasets. We expect the results to be available in the coming one or two days. Once the manuscript is updated to include these results we will report them here.
>
> **Re: What does X% IG mean? Is a model with 90% IG good enough?**: Similar to $R^2$ or correlation, this depends a lot on the context. Depending on the GS model, an IG of 90% could be considered a good model. This is why it is so important to come up with the best GS model possible which is the focus of the current paper. As opposed to correlation-based metrics which only care about the mean, for a 90% IG the encoding model needs to capture the entire response distribution well. Importantly, IG allows us to investigate which aspects of the distribution the model does not capture well by carefully matching different moments of the data and the model distribution (see e.g. our analysis in Fig. 4c), which could guide further development of the model to close the gap between the trained model and the GS model. Therefore, whether a 90% IG is good enough or not, depends on what parts of the response distribution are desired to be accurately captured by the encoding model.
>
> **Re: Can IG be used to compare models with different distribution families?**: Yes,  IG is a rescaled likelihood measure that can be used to compare models from different families. For a valid comparison of models, however, it is important that the same Null and GS models are used.
>
> **Re: Measuring the goodness of fit via decoding**: Decoding is classically used as a performance measure when the stimulus is “simple” so we can estimate the mutual information between the original and decoded stimulus, which yields a lower bound to the information between stimulus and neural responses via the data processing inequality. However, in our case, this is not feasible since we would need to estimate mutual information between natural images, which is highly non-trivial.
> Alternatively, one could decode natural images from the neural responses and measure the performance based on that. However, this approach has two difficulties: 1) The choice of loss function on natural images is not straightforward. E.g. squared loss on pixels is highly sensitive to fluctuations that are perceptually nonsignificant (i.e. shifts by a few pixels). For the mouse, it is even less clear what a perceptually meaningful loss would be. 2) If one uses a loss on natural images, the quality can depend a lot on the amount of “image prior” built into the decoding model. However, that prior information is completely independent of the neural responses and thus does not necessarily reflect the quality of the encoding model.
> Thus we focus on measuring the performance of encoding models based on what they are supposed to do: Predict neural response (distributions) well.
>
> **Re: Comparing IG with other methods**: In the original manuscript (section 4), we compared IG with other metrics (e.g. correlation and FEVE) that are commonly used to evaluate neural encoding models. In addition, we analytically derived under what assumptions IG is equivalent to other metrics (Appendix F and G).

---

> > ### Author Response · Authors · 2022-11-17
> > **Follow up for analysis on more datasets**
> >
> > **Re: Test IG on more datasets**:  We performed the analysis you asked for using five additional datasets. The results are added in Appendix J. For a more detailed description of the results please see our general response above.

---

> ### Author Response · Authors · 2022-11-29
> **We would be happy to hear your feedback**
>
> We hope that we were able to address all your concerns. We would be happy to hear your feedback and provide more clarification if necessary.

---

> > ### Comment · Reviewer_ALKJ · 2022-11-30
> > **Response to authors' rebuttal**
> >
> > I thank the authors for their additional experiments and responses to my questions. I have some follow up questions.
> >
> > 1. About using normalized IG to compare models that are trained on different datasets. In Fig. S3, dataset 2-6 have similar unnormalized IG and normalized IG, but the rank based on unnormalized IG & normalized IG are totally different. I suspect this comparison seems sensitive to the choice of null and GS models?
> >
> > 2. About the high IG meaning. How should one distinguish a high IG is due to a good encoding model or a sub-optimal GS? Reviewer 1J5x also mentioned something similar in their review.
> >
> > 3. About using IG to compare models with different distribution families. The authors mentioned that the same null and GS models need to be used for valid IG comparison. However, if trying different distributions for neural responses, e.g. discrete Poisson distribution & continuous ZIL distribution, the null & GS models would also be different. Can you clarify more on how to use same null & GS models on different response distributions?

---

> > > ### Author Response · Authors · 2022-12-02
> > > **Addressing follow up questions**
> > >
> > > Thank you for your response. We performed additional analysis and provide further explanations to address your remaining concerns:
> > >
> > > **Re 1&2 (Rank of models different when using normalized and unnormalized IG) and (High IG from good encoding model or bad GS model?)** Generally, there could be two sources for the different rank order between IG and NInGa: i) differences in the quality of the GS model or ii) differences in the quality of the data. Regarding i): Given a choice of distribution for the GS model, the GS model is likely very close to the best upper bound that can be found since we optimize the prior hyperparameters. Regarding ii) a model in IG could appear better because it has a higher log-likelihood, but the dataset might just be easier to predict. Different datasets can exhibit different levels of signal-to-noise-ratio (SNR) which can influence how good the encoding, Null and GS models can be. By normalizing the nominator in the IG with the denominator (containing the GS model), our goal is to reduce this confounding factor. To demonstrate that the source for the differences is likely the data quality, we computed the signal-to-noise ratio  of each dataset and correlated it with IG and NInGa. The correlation between dataset SNR and NInGa evaluates to $-0.08$ while the correlation between IG and dataset SNR  is much higher at $0.21$. Therefore, the rank of the models in the plot where IG is not normalized does not correctly reflect the performance of the models w.r.t. what’s possible on that dataset.
> > >
> > > **Re 3 (Comparing models from different distribution families):** Comparing models from different continuous distribution families is generally possible since likelihood can be used for model comparison. Even the Null and GS models do not need to be from the same family. However, independent of our work, continuous and discrete distributions cannot directly be compared since one provides density values while the other provides values from probability mass functions. However, one way to make them comparable is to turn the discrete pmf into a piecewise constant density.

---

> > > > ### Comment · Reviewer_ALKJ · 2022-12-03
> > > > **Response to authors' rebuttal**
> > > >
> > > > I thank the authors for their responses. They have addressed my concerns, and I'm willing to raise my point to marginal accept.
> > > >
> > > > It would be great if the authors could put the further analysis on SNR and NInGa they did in the appendix, and add a bit of clarification on this sentence on page2:
> > > > "Therefore, it can be used for model comparison across different distribution families as long as the GS and Null model are kept the same." This current statement might cause confusion, as the authors pointed out that if comparing different distribution families, the GS and Null models might not be from the same family.

---

### Official Review · Reviewer_rHRG · 2022-11-01

**Confidence:** 2
**Correctness:** 3
**Technical Novelty And Significance:** 3
**Empirical Novelty And Significance:** 3
**Recommendation:** 6

**Clarity, Quality, Novelty And Reproducibility:**

The paper is largely written clearly and appears technically sound. The work identifies an issue in an existing metric and proposes a new approach to help with this issue.

Some typos/clarifications:

- prove -> proven in paragraph above section 2.
- Appendix 4 -> Appendix D under equation 1.
- What is f in your notation? This should be clarified.
- What is a.u. in figure 3?
- Brackets around exponentiated equation at the bottom of page 12.

**Strength And Weaknesses:**

Strengths:
1. The problem the paper is tackling is largely well motivated and seems to be of importance in neural encoding models (the reviewer is not an expert in this area)
2. The proposed Bayesian oracle solution is simple and appears quite effective empirically at reducing the issues faced by the PE oracle.

Weaknesses:
1. The reviewer thinks some aspects of the approach could be developed more thoroughly: for example, does we need the full Bayesian posterior predictive in order to obtain better upper bounds or would say using a maximum-a-posteriori PE suffice in ameliorating the issues associated to current PEs. Equivalently, are current PE oracles underperforming because they are PEs, or is it because perhaps that they are overfit to the observed repeated responses (and could be improved with regularisation)? The paragraph above section 2.2 highlighting the issues with the PE approach did not read convincingly and the suggested experiment would help if I have understood correctly.
2. Likewise, it would be interesting to see *which neurons* are the ones who are being helped by the proposed approaches in the paper: for example, one could plot the difference between the PE and Bayesian oracle as a function of the sparsity of the neuron response q, which would be interesting to see.
3. The Bayesian oracle does still seem to have a (small) proportion of neurons for which the null model outperforms the GS model (in figure 1). Is there a reason for this/can we identify which neurons these are?
4. The final section 4 seems a bit orthogonal to the rest of the paper, which is focused on improving IG. It would be good to develop why we care about IG relative to other metrics more though, but this should come earlier in the paper.
5. Could the authors provide some more motivation for what benefits does improving the upper bound for the IG evaluation metric provide in practice, e.g. can we say something about neuron encoding models that we previously couldn't with the PE upper bound?

**Summary Of The Paper:**

This paper identifies an issue in the Information Gain (IG) evaluation metric for neuronal population response prediction, whereby the upper bound 'oracle' likelihood estimate using a Point Estimate (PE) approach (i.e. using a point estimate for the parameters of some simple parameterised distribution like a zero-inflated LogNormal) is often lower than the corresponding lower bound 'null' model estimate. This issue is particularly apparent in settings of few samples, sparse responses or low signal-to-noise ratios (SNRs). The paper proposes to improve the PE approach by instead using a full Bayesian treatment to obtain the posterior predictive distribution for the 'oracle' by marginalising out the parameters with respect to their posterior distribution.

The paper demonstrates how to efficiently estimate the Bayesian 'oracle' with zero-inflated distributions (which are state-of-the-art ways to model neuronal population responses) and then demonstrates that the Bayesian oracle outperforms its PE counterpart, across different SNRs and sample sizes for the leave-one-out calculations (Figure 3). In section 3.2, the paper uses its proposed IG metric (with Bayesian oracle) to evaluate a trained neural encoding model, highlighting the importance of prior choice, and finally the compares IG and other metrics that have been used to evaluate neuronal population response in section 4.



**Summary Of The Review:**

I am recommending weak accept as I think the paper identifies and answers a clear problem, and that the proposed approach appears to be effective in practice. I'd like to see more development as to the reason why the proposed approach works (in 'weaknesses', along with other suggestions which will hopefully improve the paper).

---

> ### Author Response · Authors · 2022-11-15
> **Authors response to Reviewer rHRG (part 2)**
>
> **Re: Typos/clarifications**: Thanks for pointing these out. “a.u.” stands for “arbitrary unit” but we have removed it from the updated manuscript. And “f” in the point estimate section refers to a generic function that outputs the parameters of the distribution over a target repeat, given the responses of the other repeats. In our case, this function represents moment matching. We fixed and clarified these in the paper.

---

> > ### Comment · Reviewer_rHRG · 2022-11-21
> > **Thanks for the author response**
> >
> > Thanks for the response and for running the cases I mentioned in my review. The gap between the MAP and full posterior predictive in Figure S2e is very small (0.396 vs 0.402) and leads me to ask if it's statistically significant, can you provide error bars over the repeats/stimuli/neurons? I think if so this would need some revising of the writing on the criticisms of the PE upper bound for the motivation of the Bayesian posterior, as the MAP is a PE too. I am still unconvinced by Section 4 (as a whole) and believe it is somewhat orthogonal to the rest of the paper: the main paper atm is focused on improving the upper bound estimates for IG, whereas section 4 is describing how IG relates to other metrics, and doesn't contribute to the main focus of the paper imo (and is arguably a distinct/orthogonal contribution). I would personally recommend moving section 4 to the appendix and present a more thorough experimental analysis of the methods in the last few pages of the main paper (mirroring my initial review as well as the other reviewer's comments).
> >
> > From reading the author response and the other reviews I am keeping my score. Thanks again.

---

> > > ### Author Response · Authors · 2022-11-22
> > > **Addressing remaining concerns**
> > >
> > > Thank you for your response and for elaborating on the points that keep you from raising your score. We think that we can address all of them:
> > >
> > > **Is PP > MAP significant?** The gap between the full posterior predictive and the MAP in Figure S2e is significant. The error bars you are suggesting are actually present in the Figure. However, they are too small to be visually noticeable (except for the blue bar). The two upper bound estimates evaluate to $\textrm{MAP}=0.396 \pm 0.002$ and $\textrm{PP}=0.402 \pm 0.002$ where the error is SEM. In the final version of the paper, we will add a note about the presence of the error bars in the caption of Fig. S2e.
> > >
> > > **Relevance of Section 4**: We respectfully disagree with you on the relevance of Section 4. The point of our paper is to set the ground for neural model evaluation on a full likelihood basis and to convince the field of this evaluation measure. We believe that a convincing argument for adopting a new metric, requires answers to three questions:
> > > 1. **How do we compute this metric?** This is what Section 2 (theory) and 3.1 (experiments) are about.
> > > 2. **What are the results when applied in a real scenario?** This is Section 3.2 (experiments).
> > > 3. **How does it compare to the common metrics we have been using so far?** This is what Section 4 (theory and experiments) is about. The relevance of this last point was also pointed out by reviewer **ALKJ**, who asked: “ I wonder if the authors have done comparisons with other measures? Is IG consistent with other measures?” (We pointed him/her to Section 4).
> > >
> > > **More thorough experimental analysis of the method instead of Section 4**: As described above, we would like to keep Section 4 about the comparison to other metrics and will therefore not have this space for a more extensive experimental analysis. However, this does not mean that such an analysis has not been done. In fact, we have performed several important analyses, including:
> > > - Which distribution parameters profit the most from the PP (main paper, Fig. 3a)
> > > - Robustness to different dataset sizes (main paper, Fig. 3b)
> > > - Robustness to different levels of SNR (main paper, Fig. 3c)
> > > - Relevance of the prior hyperparameters (main paper, Fig. 4a)
> > > - In which cases does the PP outperform the PE (Appendix D, Fig. S1, proposed by you)
> > > - Is the MAP sufficient (Appendix I.1, Fig. S2, proposed by you)
> > > - Would a mixture model between PP and Null model perform better (Appendix I.2, Fig. S2, proposed by reviewer **1J5x**)
> > > - Do the results hold over multiple datasets (Appendix J, Fig. S3, proposed by reviewer **ALKJ**)
> > >
> > > These points fully cover all analyses suggested by all three reviewers with the exception of reviewer **1J5x** suggestion of comparing different encoding models. In short, we did not perform such an analysis because that was not our focus in this paper (please refer to our response to reviewer **1J5x**’s comment **Too focused on Gold Standard and lack of encoding model comparison** for a more detailed response). An additional reason is that the IG metric is best used for models that predict full response distribution and thus need a likelihood-based evaluation approach. However, common predictive models only predict a single value (e.g. mean) and not a full distribution. To compare to those models (e.g. DiCarlo/Yamins as proposed by the reviewer **1J5x**), we would need to augment these models with a response distribution, which we think not only distracts from the focus of this paper but also, in our opinion, is less important than what section 4 currently contains. Finally, pretrained models have previously been reported to underperform on mouse data [1] which is another reason why it might not be a good idea to compare these models here given that our dataset is from mouse V1.
> > >
> > > [1] Cadena, Santiago A., et al. "How well do deep neural networks trained on object recognition characterize the mouse visual system?." (2019).

---

> ### Author Response · Authors · 2022-11-15
> **Authors response to Reviewer rHRG (part 1)**
>
> Thank you for your helpful and positive feedback. We improved the manuscript (the submission is already updated and uploaded) with the addition of several analyses addressing some of your concerns. Below is a detailed response to the points raised in your review:
>
> **Re: PE using MAP to avoid overfitting**: We agree with you that the reason behind the low performance of the current PE is because of overfitting, as we have also pointed out in the paper. This was indeed the reason behind using the posterior predictive which regularizes the PE approach but also directly results in a distribution over the target response. To address your comment we implemented the MAP estimate for the GS model. Our results (Fig. S2) show that the MAP formulation does improve beyond the PE approach and comes close to the Posterior Predictive but does not exceed it. We also added the derivation for the MAP GS for the Zero-Inflated LogNormal distribution in Appendix I.1.
>
> **Re: Which neurons benefit from the Bayesian approach?**: We agree that this is an interesting question. We added a more detailed analysis on why and in which cases the PE approach fails compared to the Bayesian approach in Appendix D.
> In summary: Since Bayesian and PE approaches are identical for the zero part (i.e. they have the same value for the zero-threshold $\tau$), the difference must therefore lie in the positive part of the distribution. We identified that it is not single neurons that do or do not profit. Instead, all neurons have some responses where the effect is prominent. These responses all share the same pattern: a) The target response is slightly above the zero-threshold, making it an extreme value for the positive distribution. And b) the left-out responses are very sparse resulting in overfitting of the PE.
>
> **Re: For which neurons does the Null model outperform GS (Fig. 1)?**:  Fig. 1 only shows the responses to a *single presentation* of one image for all neurons to motivate the problem. The main goal of this figure was to show some examples to describe under what conditions the Null performs better than the GS, and how using the Bayesian approach can improve the upper bound in this case where we focus on a single trial. In general, the goal was to obtain a GS that is better than the null on average (across images and repeats). However, this does not guarantee that GS is better than Null for any given trial - Fig. 1 is one example of this. However, when we average the likelihoods across all repeats and images there are no cases where the Null is better than PP (Fig. S2 in Appendix I of the updated manuscript).
>
> **Re: IG relative to other metrics**: We think there might have been a misunderstanding. The final paragraph of section 4 addresses the question of whether IG (likelihood) and correlation always go in sync, i.e. whether optimizing correlation implies optimizing likelihood and vice versa, or whether the two even lead to the same solution/share the same optimum. In general, the answer is no, and we give conditions under which the two coincide. We think that this is an important point to understand since it implies that likelihood-optimized models can have an inferior correlation.
> In general, we motivate why a likelihood-based metric is desirable in the first paragraph of the Introduction section, and we motivate why there is a need for a likelihood-based metric that is normalized by an upper bound (e.g. Normalized Information Gain) in the second paragraph of the Introduction section. In summary: correlation just captures one aspect of the entire response distribution. If one is interested in that only, then correlation is sufficient. However, if one is interested in the entire distribution, then a more holistic measure (e.g. IG) needs to be used. This measure should ideally be normalized by an upper bound (oracle) to give a sense of how close the performance of a model is to the best achievable performance.
>
> **Re: Benefit(s) of improving the upper bound in practice**: As mentioned in **Re: IG relative to other metrics**, a metric should ideally be normalized by an upper bound (Oracle) to give a sense of the optimal performance in the presence of trial-to-trial fluctuations in the data. This is also done for other commonly used metrics such as correlation and fraction of explainable variance explained (FEVE), where the metric is normalized against an upper bound accounting for the noise. For information-based metrics, it is hard to obtain such an upper bound. The approach we propose here is to use a model (oracle) that has slightly more information (e.g. other trials with the same image presentation) than a normal predictive model so it can make a prediction about how well a predictive model could do. Improving the upper bound sets a higher bar for all other models - as log-likelihood can be used for across-model comparison.

---

### Author Response · Authors · 2022-11-15
**Authors general response (part 2)**

Below is a list of the analyses and experiments we performed to address the points raised by the reviewers:
1. Derivation and implementation of MAP estimator for Zero-Inflated LogNormal distribution (Reviewer **rHRG**)
2. Implementation of the mixture model (Reviewer **1J5x**)
3. Analysis of in which cases the Bayesian GS outperforms the PE (Reviewer **rHRG**)
4. Assessing IG on more datasets (Reviewer **ALKJ**)

Comment on the scores for **Technical Novelty And Significance** (Reviewer **rHRG**: 2, Reviewer **ALKJ**: 2, Reviewer **1J5x**: 2). While both IG as well as posterior predictive are established measures/techniques, we believe that our paper still contains significant technical contributions: 1) To our knowledge, we are the first to apply the posterior predictive distribution to obtain a robust oracle model for estimating an upper bound on IG. 2) We are the first to analytically derive the posterior predictive for the state-of-the-art zero-inflated mixture models. 3) We derive analytical conditions under which likelihood-based metrics and correlation-based metrics correspond to one another (e.g. yield the same optimal solution). Unfortunately, due to space limitations, many of these technical contributions are mostly in the supplementary material which is not required to be read and reviewed by the reviewers. We would like to take this chance to point the reviewers to these potentially hidden contributions when formulating their final score.

We hope that our responses address all potential concerns. Please let us know if you need any further clarifications.

---

### Author Response · Authors · 2022-11-15
**Authors general response (part 1)**

We would like to thank the reviewers for their very helpful comments and constructuve feedback on our manuscript. We were happy to see that the reviewers found our manuscript to be “clearly written” (Rev **ALKJ**) and “largely well motivated”  (Rev **rHRG**)“. Its topic was perceived as “identify[ing] a rather serious deficiency within the current state of the field” (Rev **1J5x**) and therefore being ”of importance” (Rev **rHRG**) and addressing an “important question” (Rev **ALKJ**). Also, the results were perceived as “quite effective empirically” (Rev **rHRG**).

The main points raised by reviewers were:
1. Whether and how IG can be used to compare different models and the relevance of upper bounds in this context (Reviewers **ALKJ** and **1J5x**)
2. Why do we care about IG compared to other metrics? (Reviewers **rHRG**)
3. A comparison of the Posterior Predictive upper bound estimate with other approaches such as the maximum a posteriori estimate and a mixture model between the upper and lower bound estimates) (Reviewers **rHRG** and **1J5x**)
4. An analysis of which cases benefit from the proposed Posterior Predictive approach of obtaining the upper bound as opposed to the Point Estimate approach (Reviewer **rHRG**)
5. Testing the method only on a single dataset (Reviewer **ALKJ**)

We are confident that we can address all of the above concerns. While we provide a summary here, please also refer to our more detailed responses to each individual review below.

**Regarding 1**: IG is a rescaled likelihood measure and as such can be used to compare models from different families. For a valid comparison, it is however important that the same Null and GS models are used. As for the relevance of the upper bound, Reviewer **1J5x** rightly pointed out that the upper bound is not strictly needed to compare different models - unnormalized Information Gain is theoretically sufficient. However, an upper bound is needed in two common cases: 1) Assessing whether a model has achieved its “best possible” performance for a given dataset, and 2) comparing models that are trained on different datasets.
Regarding (1), other common metrics such as correlation and fraction of explainable variance explained (FEVE) are often corrected by a noise ceiling, for instance, estimated by an oracle (GS) model, in order to get a meaningful upper bound on the performance. Thus our approach parallels common practice in correlation-based metrics. Similarly, an upper bound is also needed for (2) because different datasets can exhibit different levels of achievable performance. The hope is that normalizing the performance by the upper bound estimated for each dataset makes models trained on different datasets more comparable.

**Regarding 2**: Correlation only captures one aspect of the response distribution. If one is interested in that only, then correlation is sufficient. However, many normative theories that link first principles to neural response properties, like neural sampling or probabilistic population codes, make predictions or rely on the variability of neural activity around the mean. If we want to use neural encoding models as a quantitative underpinning for these theories, we need them to predict and be evaluated on complete response distributions. If one is interested in the entire distribution then a more holistic measure (such as IG) needs to be used. This measure should ideally be normalized by an upper bound (oracle) to give us a sense of how close the model is to the achievable performance.

**Regarding 3**: We implemented both the MAP estimator as well as the mixture model suggested by Reviewers **rHRG** and **1J5x**, respectively. Our results (Fig. S2) show while both of these approaches performed better than the PE approach, our Posterior Predictive model is still superior to both of them. Complete description and derivations of the two models are added to the manuscript in Appendix I.

**Regarding 4**: Based on Reviewer **rHRG**’s suggestion we performed a more detailed analysis on why and in which cases the Bayesian is superior to the PE approach. We identified that it is not single neurons that do or do not profit from the Bayesian approach. Instead, all neurons have some responses where the effect is prominent. These responses all share the same pattern: a) The target response is slightly above the zero-threshold, making it an extreme value for the positive distribution. And b) the left-out responses are very sparse resulting in overfitting of the PE. This extended analysis is added to the manuscript in Appendix D.

**Regarding 5**: While we addressed all other comments of all reviewers, we are still training models to produce more results on additional datasets. We expect the results to be available in the coming one or two days. Once the manuscript is updated to include these results we will report them here.

---

> ### Author Response · Authors · 2022-11-17
> **Follow up for the analysis on more datasets**
>
> **Regarding 5**: We performed an analysis similar to Fig. 4c (blue bar) but for multiple datasets. We trained the same model described in section 3.2 on five different additional datasets. Our results show that using NInGa allows a better comparison of models that are trained on different datasets which can exhibit different levels of achievable performance (Fig. S3, left vs. right). When models with the same architecture are trained on different datasets the resulting performances are more similar in the normalized IG metric (Eq. 1) than in the unnormalized IG (i.e. the numerator of Eq. 1), because the performance of the model is reported relative to the Null and Gold Standard model. In our opinion, this underlines the importance of obtaining an upper bound performance, which is the main focus of our paper.

---

### Author Response · Authors · 2022-12-05
**Summary of follow up changes**

We would like to thank all the reviewers for their constructive and helpful feedback, and we are happy that we could address their concerns and implement almost all proposed changes to our manuscript.

As a summary of the most recent round of discussion with the reviewers, we provide a list of the changes we will make to the final version of the paper:
- We will add the analysis of SNR vs IG and NInGa to Appendix J (proposed by Rev **ALKJ**)
- We will add some clarification on the choice of the distribution family for the Null and GS model (proposed by Rev **ALKJ**)

---

### Decision · Program_Chairs · 2023-01-20

**Decision:**

Accept: poster

**Justification For Why Not Higher Score:**

The reviewers noted a number of weaknesses in the paper, which suggest a higher score would not be appropriate. The empirical evaluations should be expanded; the framing/organization of the paper should be improved; the definition of the gold standard could be streamlined; etc.

**Justification For Why Not Lower Score:**

This paper identifies a significant weakness with current practice for evaluating neural encoding models, and the message presented in this paper will bring value to the community of researchers using deep network features to predict e.g. biological responses.

**Metareview: Summary, Strengths And Weaknesses:**

This paper introduces a new metric for evaluating and comparing neural encoding models based on information gain, for which a gold standard oracle model is introduced as an upper bound.

The reviewers agree that the paper and setting are well-motivated, that information gain is clearly a superior measure of performance (relative to existing conventions), and that the proposed oracle is simple and empirically effective choice. Several concerns relate to the framing of the paper, with a large focus on first principles and the Gold Standard, with less emphasis on empirical demonstrations. Ultimately, however, as one reviewer stated, "The main strength of this paper is that it identifies a rather serious deficiency within the current state of the field....it presents a very well-known and rigorous solution."  All told, the strengths outweigh the weaknesses and the paper lands just above the bar for acceptance.

**Note From Pc:**

if the above contains the word "oral" or "spotlight" please see: "oral" presentation means -> notable-top-5% and "spotlight" means -> notable-top-25%. As stated in our emails, we are disassociating presentation type from AC recommendations